# Healthcare Practitioners Knowledge of Shoulder Injury Related to Vaccine Administration (SIRVA)

**DOI:** 10.3390/vaccines10121991

**Published:** 2022-11-23

**Authors:** Laura Jane Mackenzie, Jaquelin Anne Bousie, Phillip Newman, Janique Waghorn, John Edward Cunningham, Mary-Jessimine Ann Bushell

**Affiliations:** 1Faculty of Health (Physiotherapy), University of Canberra, Bruce, ACT 2617, Australia; 2UC Research Institute for Sport and Exercise, University of Canberra, Bruce, ACT 2617, Australia; 3Department of Pharmacy, King’s College London, London WC2R 2LS, UK; 4Royal Melbourne Hospital, University of Melbourne, Parkville, VIC 3050, Australia; 5Epworth Richmond, Richmond, VIC 3121, Australia; 6Faculty of Health (Pharmacy), University of Canberra, Bruce, ACT 2617, Australia

**Keywords:** Shoulder Injuries Related to Vaccine Administration, SIRVA, iatrogenic, anatomical knowledge, immunisation, landmarking techniques

## Abstract

Background: Vaccine pharmacovigilance is at the forefront of the public eye. Shoulder Injuries Related to Vaccine Administration (SIRVA) is a poorly understood Adverse Event Following Immunisation, with iatrogenic origins. Criteria for medicolegal diagnosis of SIRVA is conflicting, current literature and educational materials are lacking, and healthcare practitioner knowledge of the condition is unknown. Methods: A cross-sectional, convenience sampled survey, utilising a validated online questionnaire assessed practitioner knowledge of SIRVA, safe injecting, and upper limb anatomy, and preferred definition for SIRVA. Results: Mean scores were moderate for safe injecting knowledge (69%), and poor for knowledge of anatomy (42%) and SIRVA (55%). Non-immunising healthcare practitioners scored significantly (*p* = 0.01, and < 0.05, respectively) higher than immunising practitioners for anatomy (2.213 ± 1.52 vs. 3.12 ± 1.50), and safe injecting knowledge (6.70 ± 1.34 vs. 7.14 ± 1.27). Only 52% of authorised vaccinators accurately selected a 40 × 20 mm area recommended for safe injecting. Majority (91.7%) of respondents thought nerve injuries should be included in the diagnostic criteria for SIRVA. Discussion and conclusions: Greater education and awareness of SIRVA is needed in all healthcare disciplines. Consensus regarding SIRVA definition is paramount for accurate reporting and improved future understanding of all aspects of SIRVA.

## 1. Introduction

As the world has recently and tragically been reminded throughout the COVID-19 pandemic, vaccines are one of the greatest achievements of modern medicine. Despite their benefits, vaccines can also present with challenges. Shoulder Injuries Related to Vaccine Administration (SIRVA), the preferred medicolegal term since 2017 for an adverse event following immunisation (AEFI) affecting the shoulder musculoskeletal region, is an uncommon and poorly understood consequence of improper vaccination administration [1,2]. An iatrogenic mechanism of injury, SIRVA is causally linked to improper vaccination technique or location, and is considered to be preventable by the Australian Immunisation Handbook [3] through anatomical landmarking techniques [4]. However, use of these preventative strategies is poor, with 97% of participants in one study self-reporting they do not use landmarking techniques [5]. This may be because description of such landmarking techniques is lacking across immunisation handbooks and best practice guidelines [6,7,8].

Evidence related to SIRVA is sparse, with the majority of understanding coming from case studies and case series. The largest and highest quality source of evidence for SIRVA is a retrospective cohort study of the Vaccine Adverse Event Reporting Scheme (VAERS), the pharmacovigilance database for the United States of America [9]. This study examined adverse event reports from influenza vaccinations from between 2010 and 2017 [9]. Analysis of the almost 60 thousand reports yielded 1220 reports suspected to be SIRVA. Based on this, an incidence rate of 1.5–2.5% was estimated by the authors [9]. However, this cannot be considered a true estimation of incidence, as the denominator used to calculate the incidence rate was number of received AEFI reports, rather than number of administered vaccines. Instead, this rate should be considered as a proportion of SIRVA cases from within all AEFI’s reported to VAERS. VAERS and other pharmacovigilance databases are also formed from self-reported or non-mandatory medical data, which can lead to a misrepresentation of actual incidence, as potential cases can be missed if individuals are not aware of the reporting process [10]. A recent review of the VAERS database yielded 305 cases since 2021 reporting the term SIRVA, with 76.3% of reports being female [11]. Under-reporting, known errors in passive reporting systems, and low clinician awareness of the condition were proposed as key limitations to estimation of incidence [11]. Education of clinicians, increased awareness, and use of preventative techniques are reported as being crucial for prevention of SIRVA [11].

Criteria for the medicolegal diagnosis of SIRVA are currently conflicting within the literature. The medical dictionary for regulatory activities utilises the American definition, which excludes neurological or nerve injuries (Table 1) [9,12]. Additionally, seen within the literature is the Australian definition, which does not directly exclude neurological injuries, and includes a suspicion of incorrect administration technique (based on patient report or clinical suspicion from either visualization of the administration site or patient description) (Table 1) [2]. While these are the most utilised, other definitions that blend or expand these two sets of criteria have also been used [13].

As a relatively young topic of interest, there is limited available research, compounded by differing diagnostic criteria and reporting of SIRVA cases as their induced conditions [16,17]. There is also differing levels of anatomical knowledge among healthcare practitioners, based on their profession and tertiary education [5,18,19]. Lastly, there is evidence of poor healthcare practitioner awareness of SIRVA [5,20]. The current study aimed to determine healthcare practitioner (both immunising and non-immunising) self-confidence and knowledge of vaccination practice, upper limb anatomy, and SIRVA. The study also aimed to establish healthcare practitioners preferred definition of SIRVA, between two available definitions.

## 2. Materials and Methods

From April to September 2022 an online questionnaire (Appendix A) developed using Qualtrics^®^ [21] was delivered to healthcare practitioners and healthcare students. Targeted nations included Australia, Canada, New Zealand, United Kingdom, and the United States of America. Targeted healthcare professions followed the Australian Health Practitioner Regulation Agency (AHPRA) [22] discipline list, with other professions included on an as needed basis secondary to the international nature of data collection. Face and content validation of the survey was performed in two stages with six healthcare practitioners, representing the disciplines of Medicine, Physiotherapy, Nursing and Midwifery, Osteopathy, and Pharmacy. Stage one involved evaluation of questions to ensure that the study aims were adequately examined within the survey. Healthcare practitioners were asked to consider the appropriateness of each question for their specific discipline, as the survey was to be presented to both vaccinators and non-vaccinators, delivered in a split question approach. Stage two involved healthcare practitioners experienced in the design and methodology of survey-based studies. These practitioners examined question construction to ensure questions were not leading, confusing, or could have different interpretations.

Survey distribution was performed via social media outlets such as Facebook and Twitter, direct emails to regulatory or educational bodies, and through profession specific newsletters. Using these outlets, promotional material including a link and QR code were presented to prospective participants. Data were collected via convenience sampling with voluntary participation. A sample size of 400 participants was determined using Slovin’s formula, with the population size of N being the approximate number of registered healthcare practitioners from the target nations, and a margin of error set at 0.05 or 5% [23]. Participants were included if they were previously, currently, or in training to be a healthcare professional or authorised injector for human immunisation purposes. Participants were excluded if they were unwilling or unable to provide informed consent, or if they were not a healthcare professional or healthcare student. Individuals who fulfilled the inclusion criteria and provided informed consent for participation were granted access to the online survey. Ethics approval was obtained from University of Canberra Human Research Ethics Committee for a high-risk project involving humans on 10 March 2022, project number 9192. An amendment following validation was approved on 12 April 2022.

### Questionnaire Design

Eight demographic questions asked about country of training (healthcare degree and specific immunisation training) and practice, time spent in vaccination practise (when applicable), and whether participants were students or qualified, and from a vaccinating or non-vaccinating profession. Participants knowledge, beliefs, behaviours, and self-assessed confidence surrounding safe injecting practices were examined through 24 items including a seven-point Likert scale, click all that apply, and image-based questions (Figure 1).

Knowledge of upper limb anatomy was assessed using nine model image items. Two Likert style questions examined participant self-confidence of knowledge of upper limb anatomy (Figure 2). To reduce potential confounding variables, a singular standardised image was used with a low body weight model so that anatomical structures were clearly visible.

Knowledge and self-assessed confidence specific to SIRVA was assessed through eight items which examined causes, induced conditions, and at-risk structures, through click all that apply and Likert style questions (Figure 2). Participants’ preferred definition of SIRVA was assessed using two items (Figure 3 and Figure 4).

All data were assessed using SPSS version 28.0 (IBM, New York) [24]. Descriptive statistics were used to ascertain sample frequencies for each survey question, and independent samples *t*-tests were performed for group comparisons with a *p*-value set at <0.05.

## 3. Results

A total of 225 individuals participated in the online survey, with 175 answering all questions presented. Key participant demographics included: 60.9% of participants nominating Australia as their country of practice, 72.4% being fully qualified within their profession, and 59.6% being authorised vaccinators or immunisers (Table A2). Respondents represented all targeted disciplines registered under the Australian Health Practitioner Regulation Agency (AHPRA).

### 3.1. Knowledge of Upper Limb Anatomy

Participant self-assessment or “confidence” of knowledge was high for shoulder anatomy (85.8%) and safe injecting practices (78.7%), with moderate confidence noted for SIRVA knowledge (65.8%). Participants from an immunising profession were more confident in their ability to report a suspected case of SIRVA to a pharmacovigilance body than participants from non-immunising healthcare professions (76% vs. 42%) with similar findings noted for referral of suspected cases to appropriate healthcare disciplines (80% vs. 46%). No statistical comparisons were made for the above and examination was limited to descriptive statistics, secondary to varied sample sizes for each discipline. Mean scores were moderate for safe injecting knowledge (6.86 ± 1.33, maximum achievable score of 10), and poor for knowledge of anatomy (2.54 ± 1.57, maximum achievable score of 6) and SIRVA (19.95 ± 5.31, maximum achievable score of 36) (Table 2).

Using data obtained from participants clicking on where they believed specific anatomical structures were located on images of the shoulder, binary (correct vs. incorrect) means for anatomical knowledge were calculated. Participant responses were considered “correct” if they fell within the acceptable distance from the target (with size ranging from 30 mm × 30 mm to 80 mm × 60 mm) for each anatomical structure (Table 3), with the target location determined by the validation team using an anatomical textbook [25]. Sizes of the “correct” zones for the binary calculation were intentionally larger than the target anatomical structure to allow for leniency and to account for differences between presented images on a variety of electronic devices. When separated into authorised immunisers and non-immunisers, binary accuracy for the deltoid immunisation site had large variation between groups (54% correct vs. 84% correct, respectively). Anatomical knowledge accuracy scores were highest for the deltoid immunisation site and the subacromial/subdeltoid bursae, with lowest means noted for the axillary and radial nerves (Table 3). Average selected distance (pixel distance scaled to the live model, using a pixel to mm ratio of 1:0.63) from targeted anatomical structure ranged from 0.3 mm to 54 mm (*X* axis) and 11.51 mm to 79.49 mm (*Y* axis) (Table 3). Average mm distance for each profession from deltoid immunisation site target is presented in Appendix B (Figure A1).

Heat map visualisations of healthcare practitioner selected locations of anatomical structures (Figure 5) and target locations (Figure 6) are presented below.

When separated into healthcare professions, greatest anatomical knowledge mean scores were seen in the disciplines of Physiotherapy and Pharmacy (Table 4).

### 3.2. Knowledge of Safe Injecting Practices

Participants had greatest accuracy in identifying “incorrect” examples of immunisation administration (mean scores = 0.83, 0.90, 0.82) (Table 5). Lowest mean accuracy was noted for examples in which the model’s hand was placed on their hip (Image 3 = 0.42, Image 7 = 0.44) (Table 5).

Immunising healthcare practitioners largely self-reported as using anatomical landmarking techniques (76.5%), however, only 45 participants (39.1%) self-reported as using these techniques every time. All healthcare practitioners (both immunising and non-immunising) were asked to consider benefits and limitations of landmarking techniques. Main benefits of landmarking techniques were noted as “providing an accurate site for injection” (selected by 78.5% of respondents) and “protecting important underlying structures from being damaged” (selected by 75% of respondents) (Table 6). Other benefits were described by eight participants and included “reducing pain”, “conforming to best practice”, “standardisation”, and “improved effectiveness of immune response”.

Main limitations of landmarking were noted as “time consuming” (selected by 28.5% of respondents) and “inaccurate” (selected by 17% of respondents) (Table 7). Other limitations were described by 14 participants and included “patient discomfort with undressing”, “patient belief that landmarking technique may be site of injection”, “unable to account for muscle mass variations”, “changing technique preference by regulatory bodies”, “physical characteristics of practitioner e.g., hand size” and “anatomical variations”.

### 3.3. Group Comparisons for Knowledge Domains

Independent samples *t*-tests were performed for knowledge scores of anatomy, safe injecting practices, and SIRVA, comparing immunising and non-immunising professions, fully qualified practitioners and students, and self-declared high and low confidence (Table A3, Table A4 and Table A5). Non-immunising professions on average scored higher than immunising professions in all three knowledge assessments (Table A2). Statistical significance was found between immunising and non-immunising professions for anatomy knowledge (2.213 ± 1.52 vs. 3.12 ± 1.50, *p* ≤ 0.001, 95% CI = −1.38 to −0.43) and safe injecting knowledge (6.70 ± 1.34 vs. 7.14 ± 1.27, *p* = 0.033, 95% CI = −0.86 to −0.04) (Table A2 and Table A3). Fully qualified practitioners had higher mean scores than students for anatomy and SIRVA knowledge with significance achieved for the latter (20.40 ± 5.13 vs. 18.50 ± 5.68, *p* = 0.037, 95% CI = −0.11 to 3.70), with students scoring higher for safe injecting knowledge (Table A4). Those who were “confident” in their knowledge scored higher in anatomy and SIRVA than those “not confident” (Table A5). However, the inverse was noted for safe injecting knowledge, where those “not confident” scored higher than practitioners who were “confident” (6.70 ± 1.35 vs. 7.47 ± 1.16, *p* ≤ 0.05, 95% CI = 0.26 to 1.28) (Table A5). When separated into respondents who were immunisers and non-immunisers, immunising respondents had minimal differences in confidence of safe injecting practice, with a wide variety in knowledge scores for safe injecting practice (Figure 7). Non-immunising healthcare practitioners maintained an inverse relationship between confidence and knowledge of safe injecting practices (Figure 7).

### 3.4. Preferred Diagnostic Criteria

No clear preference was identified for a definition of SIRVA, with 41.4% of participants selecting the option for both (USA/MedDRA vs. AUS) (Table 8). However, 91.7% of the 175 respondents to the final question indicated that nerve injuries should be included in the definition of SIRVA (Table 8).

## 4. Discussion

The current study indicates there is a low level of healthcare practitioner and healthcare student knowledge related to SIRVA, shoulder anatomy, and safe injecting practices. To date, only one other study, conducted in 2005, has explored this area [5]. Like the current study, findings of that study found practitioners had moderate confidence in their anatomy and SIRVA knowledge, with poor actual knowledge of which structures were at risk [5]. Differing levels of human anatomy tertiary education was suggested as the primary reason for this finding [5]. From the current study, large variation of selected location was noted for the anatomical landmarks of the acromion process, axillary nerve, radial nerve, and deltoid tuberosity (Table 3, and Figure 5). This is concerning as, not only are these structures at risk of injury, but the acromion process and deltoid tuberosity are also the main bony landmarks used in the anatomical landmarking techniques described for human immunisation [20,26,27].

The findings of the current study indicate that the disciplines of Physiotherapy and Pharmacy have the greatest levels of anatomical knowledge related to the shoulder. Physiotherapy places a large focus on surface anatomy and anatomical knowledge within qualifying degrees [28]. The discipline of Pharmacy experiences high levels of scrutiny and specific training for pharmacist vaccinators (particularly in countries such as Australia where this discipline is relatively new to immunisation practice) [29]. These may be possible reasons behind the findings of the current study. However, mean scores of anatomical knowledge, even for these highest performing professions, indicate that knowledge is poor. Anatomical knowledge declines in healthcare professions, even while undertaking study, but more so in the first two years following graduation [28,30,31]. Development and integration of anatomical revision materials has been suggested as a method to reduce this knowledge attrition [28,32]. A review of Physiotherapy and Pharmacy anatomy content may guide development of these materials, given the apparent improved knowledge retention in these disciplines.

Participants in the current study demonstrated high levels of accuracy in identifying “incorrect” examples of human immunisation technique, with moderate accuracy in identifying “correct” examples. Lowest accuracy was found for images in which the model’s hand was placed on their ipsilateral hip, a technique which has been described in the literature to both relax the deltoid muscle and move the axillary nerve closer to the acromion process [20,27,33]. However, this has not been described in immunisation handbooks, which may have caused the discrepancy [6,7,8,26]. Practitioner respondents in the McGarvey & Hooper [5] study largely self-reported not using landmarking techniques (96%), instead relying on visualisation of the target muscle. Unclear policy and poorly regulated human immunisation training was the main suggestion for this finding [5]. However, in numerous education pieces related to the prevention of SIRVA, complacency of practitioners has been suggested [20,27]. Self-reported use of landmarking techniques in the current study was far higher than those in the McGarvey& Hooper [5] study (39.1% strongly agree as always using techniques), however, still significantly below the 100% use recommended by the Australian Immunisation Handbook [3,26]. A number of respondents in the current study (29%) perceived landmarking techniques as being time consuming, identifying this as a limitation to use. This raises concerns regarding actual practitioner application of these techniques, given most are designed to be implemented while the immunisation is being administered [3]. The Australian Immunisation Handbook appears to be the only resource to discuss anatomical landmarking techniques, with other handbooks and guidelines only recommending needle length, angle of delivery, and site of injection [6,7,8,26]. Inclusion of anatomical landmarking techniques within immunisation guidelines has potential to improve use of these protective strategies by healthcare practitioners.Reasons behind poor knowledge of SIRVA may be related to differing diagnostic criteria used between practitioners and governing bodies, limited availably of educational materials, and under-reporting of suspected cases leading to an inaccurate measure of incidence [17]. This study highlights a need for greater education related to pharmacovigilance bodies and reporting pathways available to healthcare practitioners. Only 42% of non-immunising healthcare practitioner respondents felt confident to report a suspected case of SIRVA to the relevant governing body. Non-immunising healthcare disciplines, while not involved in vaccine administration outside of surge workforces, have the potential to be first contact practitioners/disciplines for patients who have suffered from a SIRVA and are likely to diagnose and manage SIRVA induced conditions. As such, if not reported prior, first contact practitioners should report suspected cases to pharmacovigilance bodies. However, the authors do recognise that reporting of AEFIs is mandatory only for pharmaceutical companies, not for healthcare practitioners, in Australia, the United Kingdom, the United States of America, Canada, and New Zealand, which does diminish the likelihood of understanding prevalence with confidence [7,8,14,26,34,35]. Practitioner respondents to the survey had confidence levels greatly exceeding their actual knowledge (Figure 7). As such, the development of educational materials alone may not be sufficient in addressing the low knowledge base of practitioners. It is probable that practitioners with high self-confidence are unlikely to identify knowledge gaps and seek out appropriate additional training. As such, the authors propose that educational materials for SIRVA should be included in tertiary educational courses, with mandatory update training for already qualified practitioners.

Given the conflicting diagnostic criteria, and the overwhelming preference for neurological injuries to be included in the SIRVA definition, the initial estimated proportion of 1.5–2.5% by Hibbs et al. [9] utilising the American definition, is brought into question. This exclusion of neurological injuries has potentially lowered the estimated proportion and further reduces the awareness and understanding of the condition. Improper immunisation has been demonstrated as having a causal link to neurological injuries through numerous retrospective cohort studies, yet this area has been discounted and excluded [36,37,38]. Using the proposed estimated proportion of 1.5–2.5% and applying it to 2021 COVID-19 AEFI reports in Australia would give 1461 to 2435 cases [39]. When expanded to global reports of AEFI’s, using the World Health Organisation’s Vigiaccess database, COVID-19 AEFI’s in 2021 resulted in 2,878,798 reports, which would give an estimated proportion of 43,181 to 71,969 cases as SIRVA [40]. A more encompassing diagnosis that includes neurological injuries would likely greatly increase this figure. Moving forward, to reduce confusion and aid in accurate reporting, a consensus definition that addresses the inclusion of nerve injuries should be developed and implemented globally.

## 5. Conclusions

The findings of the current study suggest that healthcare practitioners’ knowledge of SIRVA, shoulder anatomy, and correct immunisation techniques are poor. A lack of knowledge of the underlying anatomy and safe injecting practices, increases risk of errors during administration. Lack of knowledge of the condition itself reduces ability to recognise, diagnose, manage, and report suspected cases. SIRVA is of concern to all healthcare practitioners. Greater education, awareness, and consensus regarding SIRVA definition is paramount to reducing the incidence of this challenging mechanism of injury.

### 5.1. Limitations

This study is limited by its sample size not reaching the level of statistical power, which does not allow for wider generalisability. It is also limited by the potential for self-selection bias.

### 5.2. Clinical Relevance and Proposed Future Research

Possible future directions of this research include development of a consensus definition for SIRVA, a review and update of current immunisation education with a focus on available landmarking techniques and preventative strategies, refreshing of human anatomy knowledge within professional development courses with potential for mandatory update and knowledge assessment, reviews of relevant undergraduate healthcare degrees, and the development of SIRVA educational materials.

## Figures and Tables

**Figure 1 vaccines-10-01991-f001:**
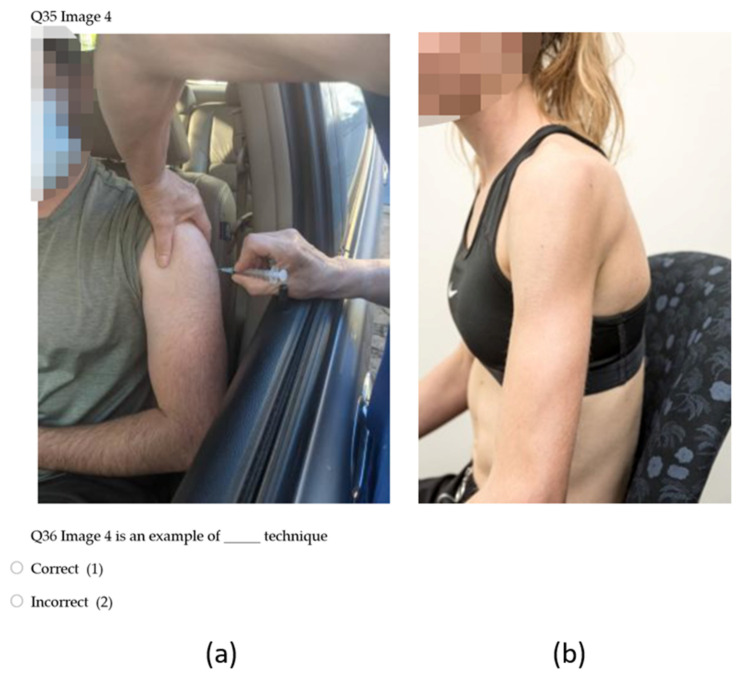
Image examples from Qualtrics^®^ questionnaire. Key: (**a**) Qualtrics question examining knowledge of safe injecting practices. (**b**) Standardised image of low body weight model.

**Figure 2 vaccines-10-01991-f002:**
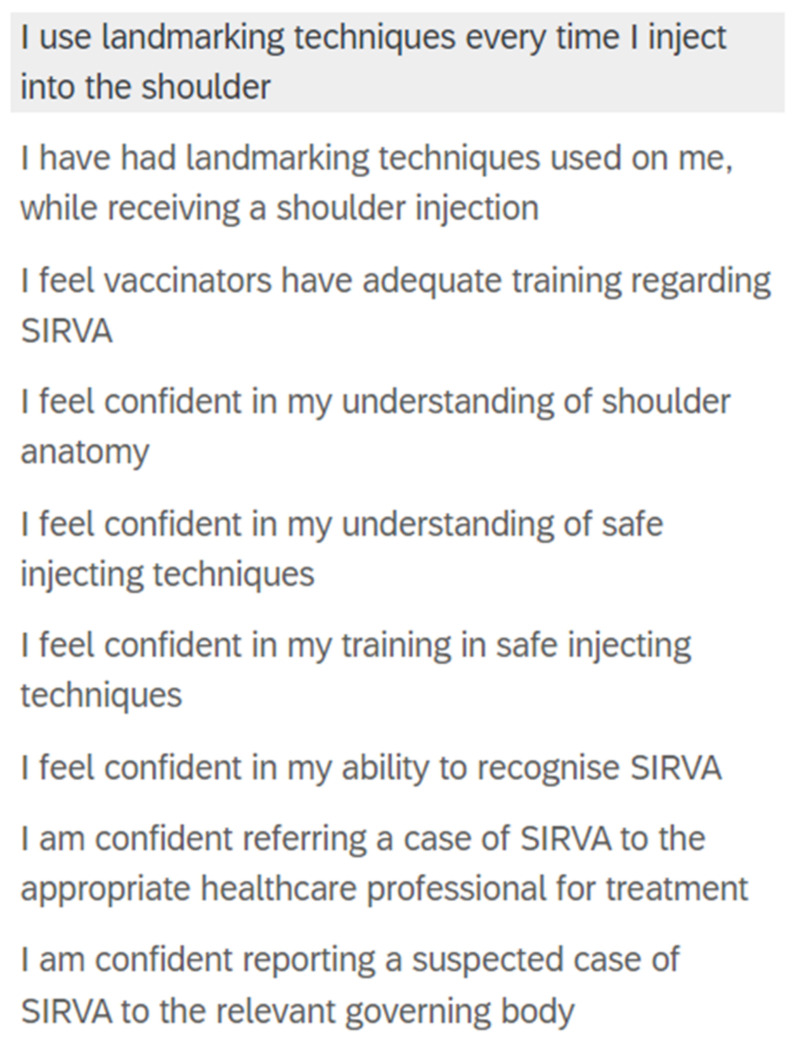
Likert scale Qualtrics question examining participant behaviours and self-confidence.

**Figure 3 vaccines-10-01991-f003:**
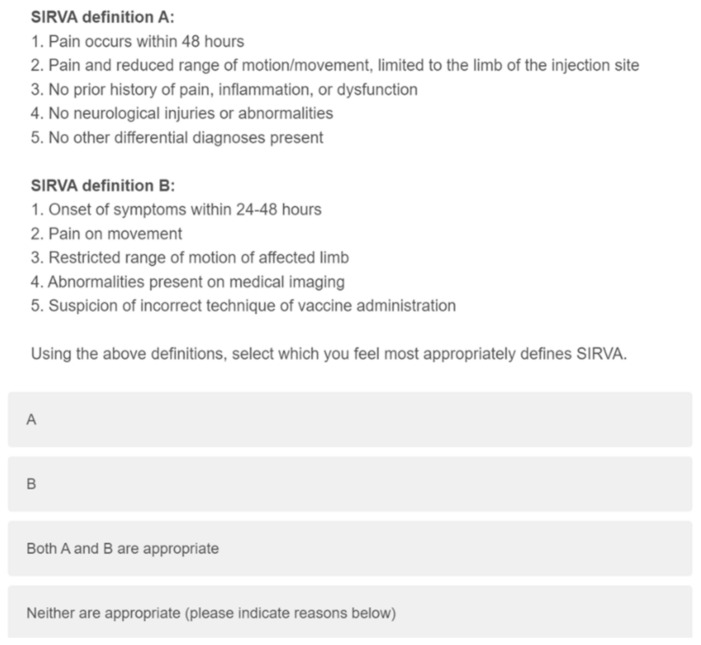
Qualtrics question examining SIRVA diagnostic criteria preference.

**Figure 4 vaccines-10-01991-f004:**
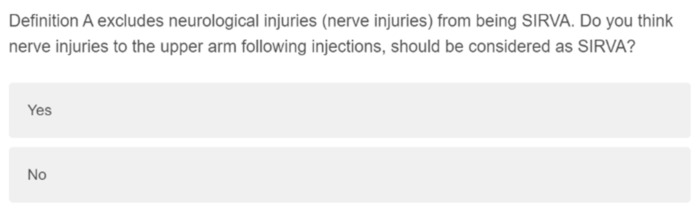
Qualtrics question examining SIRVA diagnostic criteria, specific to neurological injuries.

**Figure 5 vaccines-10-01991-f005:**
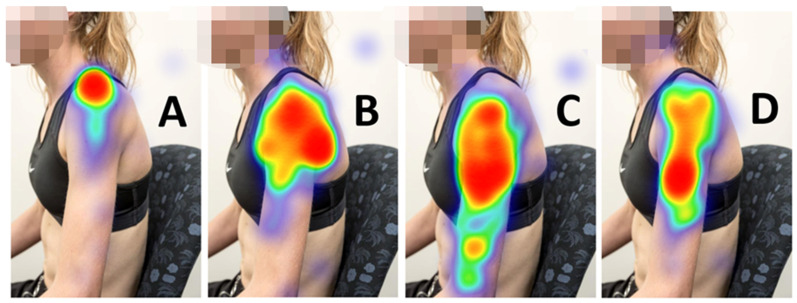
Heat map visualisation of selected location of anatomical structures. Key: (**A**) = Acromion process; (**B**) = Axillary nerve; (**C**) = Radial nerve; (**D**) = Deltoid tuberosity.

**Figure 6 vaccines-10-01991-f006:**
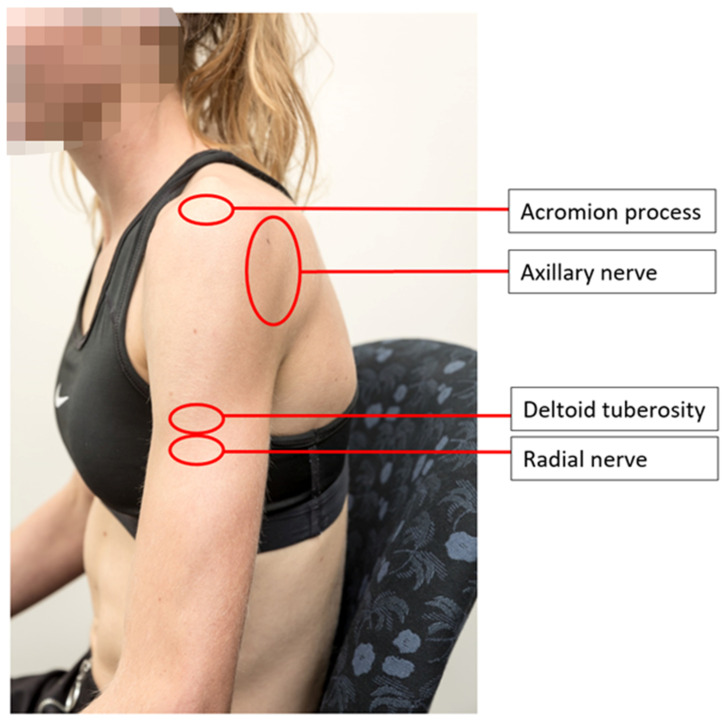
Locations of anatomical structures.

**Figure 7 vaccines-10-01991-f007:**
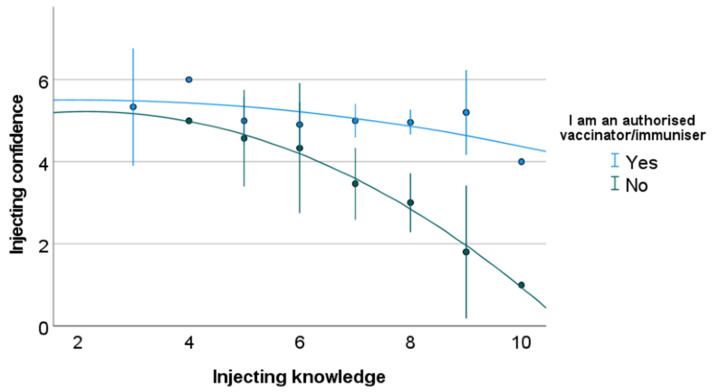
Non-linear correlation between confidence and knowledge of safe injecting practices, between authorised immunisers and non-immunisers. Key: Error bars = 95% Confidence Intervals.

**Table 1 vaccines-10-01991-t001:** SIRVA diagnostic criteria.

USA/MedDRA [14,15]	AUS [2]
Pain occurs within 48 h	Onset of symptoms within 24–48 h
Pain and reduced range of motion limited to the limb of the injection site	Pain on movement
No prior history of pain, inflammation, or dysfunction	Restricted range of motion of affected limb
No neurological injuries or abnormalities	Abnormalities present on medical imaging
No other differential diagnoses present	Suspicion of incorrect technique of vaccine administration

Key: USA = United States of America; MedDRA = Medical Dictionary for Regulatory Activities; AUS = Australia; SIRVA = Shoulder Injuries Related to Vaccine Administration.

**Table 2 vaccines-10-01991-t002:** Pooled knowledge scores for shoulder anatomy, safe injecting, and SIRVA.

	n	Minimum	Maximum	Mean	SD	Accuracy %
Anatomy knowledge	169	0.00	6.00	2.54	1.57	42
Safe injecting knowledge	172	3.00	10.00	6.86	1.33	69
SIRVA knowledge	185	3.00	32.00	19.95	5.31	55

Key: n = number of respondents; SD = Standard deviation; % = Percent; SIRVA = Shoulder Injuries Related to Vaccine Administration.

**Table 3 vaccines-10-01991-t003:** Accuracy of anatomical knowledge on a model image.

Anatomical Structure	Size of Acceptable Target Location	Proportion of Selected Correct Target	Scaled Mean mm Distance from Target		Scaled SD mm Distance from Target	
			*X* axis	*Y* axis	*X* axis	*Y* axis
Deltoid immunisation site	40 mm × 20 mm	0.66	0.39	10.17	11.51	33.71
Acromion	80 mm × 40 mm	0.37	4.92	12.96	17.21	50.84
Axillary nerve	70 mm × 30 mm	0.27	28.73	54.14	32.42	56.43
Radial nerve	30 mm × 30 mm	0.34	4.26	44.62	27.93	79.49
Deltoid tuberosity	50 mm × 30 mm	0.35	5.80	44.11	22.02	68.73
Bursae	80 mm × 60 mm	0.58	6.02	36.99	18.76	62.08

Key: Binary mean= Correct (1) vs. Incorrect (0); SD = Standard Deviation.

**Table 4 vaccines-10-01991-t004:** Mean anatomical knowledge of specific healthcare disciplines.

Healthcare Discipline	n	Mean	SD
Chinese Medicine	11	1.00	0.63
Chiropractic	13	1.08	0.95
Medicine	17	1.29	1.10
Medical Radiation Science	13	1.46	1.13
Nursing and Midwifery	55	2.51	1.39
Osteopathy	9	1.11	1.05
Paramedicine	5	1.20	0.84
Pharmacy	23	3.52	1.59
Physiotherapy/Physical therapy	44	3.52	1.25
Physician’s Assistant	4	1.75	0.50

Key: n = Number of participants; SD = Standard Deviation.

**Table 5 vaccines-10-01991-t005:** Respondent accuracy in identifying correct and incorrect examples of immunisation.

Safe Injecting Image	n Accurate	n Inaccurate	% Accurate
Image 1 (Correct)	135	37	78%
Image 2 (Correct)	106	66	62%
Image 3 (Correct)	73	99	42%
Image 4 (Correct)	105	67	61%
Image 5 (Incorrect)	143	29	83%
Image 6 (Incorrect)	154	18	90%
Image 7 (Correct)	75	97	44%
Image 8 (Incorrect)	141	31	82%
Image 9 (Correct)	134	38	78%

Key: n = Number of Participants; SD = Standard Deviation; % = Percent.

**Table 6 vaccines-10-01991-t006:** Respondent nominations of benefits of landmarking techniques.

Benefits of Landmarking	Yes	No
Provides an accurate site for injection	179	49
Protects the vaccinator from liability	88	140
Protects important underlying structures from being damaged	172	56
Improves patient trusts in the practitioner	110	118
No benefits	2	226
Other	8	-

**Table 7 vaccines-10-01991-t007:** Respondent nominations of limitations of landmarking techniques.

Limitations of Landmarking Techniques	Yes	No
Time consuming	65	163
Inaccurate	39	189
Unnecessary, eyeballing is sufficient	22	206
Not effective at protecting underlying structures	21	207
No limitations	94	134
Other	14	-

**Table 8 vaccines-10-01991-t008:** Practitioner preferred opinion of diagnostic criteria (SIRVA definition).

Preferred Definition	n	Valid %
SIRVA definition		
USA/MedDRA	50	29.6
AUS	44	26.0
Both	70	41.4
Neither	5	3.0
Missing	56	
SIRVA nerve inclusion		
Yes	155	91.7
No	14	8.3
Missing	56	

Key: n = number of respondents; % = Percent; SIRVA = Shoulder Injuries Related to Vaccine Administration; USA = United States of America; MedDRA = Medical Dictionary for Regulatory Activities; AUS = Australia.

## Data Availability

The data presented in this study are available in Appendix A.

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
