# Peer review of "Healthcare Practitioners Knowledge of Shoulder Injury Related to Vaccine Administration (SIRVA)"

_vaccines, 2022, doi:10.3390/vaccines10121991_

Round 1
Reviewer 1 Report
Thank you for the opportunity to review this paper, which presents the results of a survey of various health professionals on knowledge related to SIRVA. This work addresses a novel and important research question that requires further study, and advocates for a global definition of SIRVA. I offer the following comments/suggestions for your consideration:
Introduction:
Lines 33-37: I hesitate to use the term 'side effect' to describe SIRVA. In my opinion, a side effect is an occurrence that can happen even with proper use of a drug or procedure. Conversely, SIRVA is directly associated with improper technique. I would recommend modifying these sentences to: "Despite their benefits, vaccines can also present with challenges. Shoulder Injuries Related to Vaccine Administration (SIRVA), the preferred medicolegal term since 2017 for an adverse event following immunisation (AEFI) affecting the shoulder musculoskeletal region, is an uncommon and poorly understood consequence of improper vaccine administration."
Methods:
Line 111: There is a typo here, in the part of the sentence that says "time spent vaccinations in vaccination practice (when applicable)" - should this be time spent administering vaccinations in practice?
Discussion:
I found it interesting how in section 3.1 it was found that respondents' confidence greatly exceeded their actual knowledge. Is it therefore possible that healthcare professionals would be unable to self-identify their own knowledge gaps to seek out additional training on this? I note that you do discuss the possibility of mandatory training for injectors on this. Perhaps linking this recommendation to this finding would strengthen the argument for mandatory training.
Author Response
Reviewer 1:
Thank you for the opportunity to review this paper, which presents the results of a survey of various health professionals on knowledge related to SIRVA. This work addresses a novel and important research question that requires further study, and advocates for a global definition of SIRVA. I offer the following comments/suggestions for your consideration:
Introduction: Lines 33-37: I hesitate to use the term 'side effect' to describe SIRVA. In my opinion, a side effect is an occurrence that can happen even with proper use of a drug or procedure. Conversely, SIRVA is directly associated with improper technique. I would recommend modifying these sentences to: "Despite their benefits, vaccines can also present with challenges. Shoulder Injuries Related to Vaccine Administration (SIRVA), the preferred medicolegal term since 2017 for an adverse event following immunisation (AEFI) affecting the shoulder musculoskeletal region, is an uncommon and poorly understood consequence of improper vaccine administration."
RESPONSE: The above suggestions for lines 33-37 have been implemented, with thanks.
Methods: Line 111: There is a typo here, in the part of the sentence that says "time spent vaccinations in vaccination practice (when applicable)" - should this be time spent administering vaccinations in practice?
RESPONSE: With thanks, line 111 (now line 118) now reads: “time spent in vaccination practise (when applicable).”
Discussion: I found it interesting how in section 3.1 it was found that respondents' confidence greatly exceeded their actual knowledge. Is it therefore possible that healthcare professionals would be unable to self-identify their own knowledge gaps to seek out additional training on this? I note that you do discuss the possibility of mandatory training for injectors on this. Perhaps linking this recommendation to this finding would strengthen the argument for mandatory training.
RESPONSE: Thank you for this insightful comment. We have now added the following paragraph (Line 333-339) based on your recommendation.
“Practitioner respondents to the survey had confidence levels greatly exceeding their actual knowledge (Figure 7). As such, the development of educational materials alone may not be sufficient in addressing the low knowledge base of practitioners. It is probable that practitioners with high self-confidence are unlikely to identify knowledge gaps and seek out appropriate additional training. As such, the authors propose that educational materials for SIRVA should be included in tertiary educational courses, with mandatory update training for already qualified practitioners.”
Thank you for your review of our paper, we greatly appreciate the insightful comments you have made.
Reviewer 2 Report
This is a paper with the usual limitations of questionnaire studies based upon convenience samples recruited electronically, but the authors are aware of this, and make an appropriate comment on page 12. I agree with their view that there are a number of avenues for future research, and I think the publication of the paper would be an useful signpost for other potential researches. There are recent publications about our understanding of the nature of SIRVA that the authors should consider citing in their own work - one of them a very recent publication in Vaccine:
Bass JR, Poland GA. Shoulder injury related to vaccine administration (SIRVA) after COVID-19 vaccination. Vaccine. 2022 Aug 12;40(34):4964-4971. doi: 10.1016/j.vaccine.2022.06.002. Epub 2022 Jun 8. PMID: 35817645; PMCID: PMC9174179.
The other is a Systematic Review, carried out to acceptable PRISMA standards:
Cagle PJ Jr. Shoulder Injury after Vaccination: A Systematic Review. Rev Bras Ortop (Sao Paulo). 2021 Jun;56(3):299-306. doi: 10.1055/s-0040-1719086. Epub 2020 Dec 16. PMID: 34239193; PMCID: PMC8249056.
The second paper is published in a relatively obscure journal, but the author comes from a major North American institution.
I have a small number of specific criticisms
Line 17: The study is described in the abstract as a cross sectional survey, which is not correct. The text describes it correctly as a convenience sample, So I assume that this is an oversight
Line 62: I was puzzled by the Australian definition of SIRVA, and in particular by the idea that a suspicion of incorrect technique should be incorporated as a diagnostic criterion. It would be useful if he authors might include, perhaps as a single sentence, some comment about whose suspicion becomes a diagnostic criterion. For example, would we include a "suspicion" from a disgruntled patient?
Author Response
Reviewer 2:
This is a paper with the usual limitations of questionnaire studies based upon convenience samples recruited electronically, but the authors are aware of this, and make an appropriate comment on page 12. I agree with their view that there are a number of avenues for future research, and I think the publication of the paper would be an useful signpost for other potential researches. There are recent publications about our understanding of the nature of SIRVA that the authors should consider citing in their own work - one of them a very recent publication in Vaccine:
Bass JR, Poland GA. Shoulder injury related to vaccine administration (SIRVA) after COVID-19 vaccination. Vaccine. 2022 Aug 12;40(34):4964-4971. doi: 10.1016/j.vaccine.2022.06.002. Epub 2022 Jun 8. PMID: 35817645; PMCID: PMC9174179.
The other is a Systematic Review, carried out to acceptable PRISMA standards:
Cagle PJ Jr. Shoulder Injury after Vaccination: A Systematic Review. Rev Bras Ortop (Sao Paulo). 2021 Jun;56(3):299-306. doi: 10.1055/s-0040-1719086. Epub 2020 Dec 16. PMID: 34239193; PMCID: PMC8249056.
The second paper is published in a relatively obscure journal, but the author comes from a major North American institution.
RESPONSE: Thank you for suggesting the above references. We have added the following paragraph (beginning line 58) addressing Bass and Poland 2022:
“A recent review of the VAERS database yielded 305 cases since 2021 reporting the term SIRVA, with 76.3% of reports being female [11]. Under-reporting, known errors in passive reporting systems, and low clinician awareness of the condition were proposed as key limitations to estimation of incidence [11]. Education of clinicians and increased awareness and use of preventative techniques are reported as being crucial for prevention of SIRVA [11].”
I have a small number of specific criticisms
Line 17: The study is described in the abstract as a cross sectional survey, which is not correct. The text describes it correctly as a convenience sample, So I assume that this is an oversight
RESPONSE: Thank you for your suggestion, Line 17 has been edited for clarity and now reads: ”Methods: A cross-sectional convenience sample survey, utilising a validated online questionnaire, assessed practitioner knowledge of SIRVA, safe injecting, upper limb anatomy, and preferred definition for SIRVA.”
In this instance we have used a cross sectional design with convenience sampling methodology. https://www.ncbi.nlm.nih.gov/pmc/articles/PMC4885177/
Line 62: I was puzzled by the Australian definition of SIRVA, and in particular by the idea that a suspicion of incorrect technique should be incorporated as a diagnostic criterion. It would be useful if he authors might include, perhaps as a single sentence, some comment about whose suspicion becomes a diagnostic criterion. For example, would we include a "suspicion" from a disgruntled patient?
RESPONSE: Thank you for your comment. Lines 60-64 (now lines 68-70) now read: “Also seen within the literature is the Australian definition, which does not directly exclude neurological injuries, and includes suspicion of incorrect administration technique (based on patient report or clinical suspicion from either visualization of the administration site or patient description) (Table 1).”
While suspicion from clinicians would be preferred, the current diagnostic criteria is open to interpretation, and it allows for suspicion to first be raised by the patient. The authors propose that neither diagnostic criteria set is sufficient. It is the hope of the authors that future research and informed discussion yields a more appropriate definition.
Thank you for your review of our paper, we greatly appreciate the insightful comments you have made.